# Improvement of Cognitive Function by Fermented *Panax ginseng* C.A. Meyer Berries Extracts in an AF64A-Induced Memory Deficit Model

**DOI:** 10.3390/nu15153389

**Published:** 2023-07-30

**Authors:** Eun-Jung Yoon, Jeong-Won Ahn, Hyun-Soo Kim, Yunseo Choi, Jiwon Jeong, Seong-Soo Joo, Dongsun Park

**Affiliations:** 1Laboratory of Animal Physiology and Medicine, Department of Biology Education, Republic of Korea National University of Education, Cheongju 28173, Chungbuk, Republic of Korea; ejyoon@knue.ac.kr (E.-J.Y.); jwone@knue.ac.kr (J.J.); 2College of Life Science, Gangneung-Wonju National University, 7 Jukheon-gil, Gangneung 25457, Gangwon, Republic of Korea; 0000@gwnu.ac.kr (J.-W.A.); k4609@gwnu.ac.kr (H.-S.K.); 3Huscion MAJIC R&D Center, 331 Pangyo-ro, Seongnam 13488, Gyeonggi, Republic of Korea

**Keywords:** cognitive function, Alzheimer’s disease, ginseng berry extracts, ginsenosides, nerve growth factor, brain-derived neurotrophic factor, neural stem cells, ICR mice

## Abstract

This study investigated the potential therapeutic properties of fermented ginseng berry extract (GBE) for Alzheimer’s disease (AD). Fermented GBE was examined for its ginsenoside content and physiological properties, which have been suggested to have neuroprotective effects and improve cognitive function. The results showed that fermented GBE contains high levels of major active ginsenosides and exhibits antioxidant and acetylcholinesterase inhibitory activities. Post-fermented GBE demonstrated therapeutic potential in AF64A-induced damaged neural stem cells and an animal model of AD. These findings suggest that fermented GBE may hold promise as a candidate for developing new therapeutic interventions for memory deficits and cognitive disorders associated with AD and other neurodegenerative conditions. However, further studies are needed to evaluate the safety, tolerability, and efficacy of fermented GBE in human subjects and to determine its clinical applications. In conclusion, our study provides evidence that fermented GBE has potential as a natural product for the prevention and treatment of AD. The high levels of active ginsenosides and antioxidant and acetylcholinesterase inhibitory activities of fermented GBE suggest that it may be a promising therapeutic agent for improving cognitive function and reducing neurodegeneration.

## 1. Introduction

Alzheimer’s disease (AD) is a prevalent neurodegenerative disorder characterized by cognitive and memory impairments [1]. Key pathological features of AD include the accumulation of amyloid beta (Aβ) and tau protein, resulting in the formation of senile plaques and neurofibrillary tangles (NFTs) [2,3]. These etiological factors damage the cholinergic nervous system, leading to a loss of learning and memory [4,5,6]. AD is projected to become an increasingly significant public health issue due to the growing aging population, with estimates predicting one million cases by 2024 and three million cases by 2050 [7].

Although various therapeutic approaches are under investigation for AD, there is a need for more effective treatments and preventative measures due to an incomplete understanding of the pathophysiology [8,9,10]. Several therapeutic interventions have been explored, including acetylcholinesterase (AChE) inhibitors like donepezil and memantine [11], stem cells for cholinergic system regeneration [12], and therapeutic antibodies for causative agent removal [13]. However, their practical applications remain limited. Recently, there has been increasing interest in the application of natural extracts as potential therapeutic agents for AD. These agents include phenolic acids and flavonoids, which exhibit antioxidative [14], anti-neuroinflammatory [15], and anti-Aβ aggregation properties [16], as well as targeting tau protein [17] and cholinergic neurotransmission [18].

Various animal models have been used for drug screening in AD patients. Typical animal models of AD include spontaneously aging animals, transgenic animals, and chemically induced animal models. Among them, chemically induced animal models using amyloid beta (Aβ), cholinotoxin, and cholinergic antagonists are relatively economical. However, they have other disadvantages, such as short memory retention and sometimes requiring highly professional skills (e.g., intracerebroventricular injection). AF64A, a cholinergic toxin widely used in screening drugs for dementia treatment development, is internalized only into cholinergic neurons by the high-affinity choline transport system, causing alterations in the mRNA expression and activity of choline acetyltransferase (ChAT). This unique characteristic sets AF64A apart from other potential causes of AD [18,19,20,21]. Therefore, injection of AF64A decreases the release of acetylcholine (ACh), leading to cognitive impairments, including memory and learning deficits, which are characteristic of AD [18,22]. The AF64A-induced model stably maintains memory deficiency for 6 weeks, making it suitable for evaluating the therapeutic effects of drug candidates.

Ginseng berry extract (GBE) is a natural product that has shown potential in treating various diseases due to its ginsenoside content. Ginseng berries contain more ginsenosides than ginseng roots [23] and have been found to prevent atherogenesis [24], improve age-related diabetes [25], exhibit antioxidative effects as cosmetics [26], prevent scopolamine-induced memory deficits [27], and increase antitumor activity through regulation of the immune system [28]. Fermentation of natural products has also gained recent attention due to its ability to alter the chemical composition and biological activity of extracts [29]. Fermented foods offer several health benefits, such as antioxidant, antimicrobial, antifungal, anti-inflammatory, antidiabetic, and antiatherosclerotic activity, as components like vitamins, minerals, and biologically active peptides are altered, and non-nutrient substances are removed during fermentation [30,31,32]. Moreover, ginsenosides, including those found in fermented *Panax ginseng* extracts, are known to be absorbed in the gut when administered orally.

In this study, we hypothesized that the active ingredients would increase in fermented GBE compared to non-fermented GBE. To test this hypothesis, we compared the ginsenoside content and biological properties of fermented and non-fermented GBE. Specifically, we examined the antioxidative effects, AChE inhibitory activity, and neuroprotective effects, as these properties are known to be relevant to the pathophysiology of Alzheimer’s disease. We also investigated the therapeutic effects of fermented GBE in human neural stem cells and an animal model of AD induced by AF64A. By evaluating the ginsenoside content and biological properties of fermented GBE, we aimed to determine whether it could hold potential as a therapeutic intervention for AD.

## 2. Materials and Methods

### 2.1. Sample Preparation

The preparation of GBE in this study followed the methodology described in earlier publications [33,34]. Fresh berries of *Panax ginseng* C.A. Meyer were collected from four-year-old plants in Yeoncheon-gun, Gyeonggi-do, Republic of Korea. The harvested berries were authenticated by Prof. Joo (Biopharmaceutical Lab, College of Life Science, Gangneung-Wonju National University, Republic of Korea), and a voucher specimen was deposited at the Herbarium of the College of Natural Science, Kangwon National University, Republic of Korea (#KWNU-98802). The collected berries underwent dehydration and seven steaming cycles at 100 °C for 2 h, each followed by 24 h drying periods at 50 °C. The fourth and seventh steam-dried berry products were mixed in equal amounts and then extracted twice using 65% ethanol (10-fold volume) for 4 h at room temperature, to prepare the GBE. The ensuing ethanol extracts were filtered using Hyundai Micro No. 2 filters (Hyundai micro, Seoul, Republic of Korea), then evaporated and lyophilized to yield the pre-fermented GBE. The fermentation process involved diluting the pre-fermented GBE with distilled water to 4 brix and fermenting with *Lactobacillus plantarum* (2.0 × 10^7^ CFU/mL) for 48 h at 30 °C and 150 rpm. Post-fermentation, the extract was centrifuged at 8000× *g* for 1 h to remove any debris and *L. plantarum*. The supernatant was then collected, filtered, and lyophilized to produce the post-fermented GBE. This study involves a comparison between the pre-fermented and post-fermented GBE samples. All fermentation processes adhered to the procedures outlined in prior literature [35].

### 2.2. High-Performance Liquid Chromatography (HPLC) Analysis

The analysis of ginsenosides was conducted utilizing an Ultimate 3000 HPLC system (Thermo Fisher Scientific, Waltham, MA, USA), equipped with a diode array detector (DAD) [36]. For ginsenoside separation, a YMC-Triart C18 column (250 × 4.6 mm, 12 nm, S-5 µm, YMC, Tokyo, Japan) was employed, with the column temperature consistently maintained at 30 °C. A gradient elution program with solvent A (water) and solvent B (acetonitrile) was applied as follows: 0–10 min (20% B), 10–40 min (20–32% B), 40–55 min (32–50% B), 55–70 min (50–65% B), 70–72 min (65–90% B), 72–82 min (90% B), 82–84 min (90–20% B), and 84–90 min (20% B). This program was conducted at a flow rate of 1.6 mL/min. Detection was achieved at 203 nm, and the retention times and peak areas of the ginsenoside peaks were logged via Chromeleon software (version 7.2.10; Thermo Fisher Scientific, Waltham, MA, USA). Each peak was identified by comparing its retention times and UV spectra with those of the ginsenoside standard mixture (Biopurify Phytochemicals, Chengdu, China).

### 2.3. Determination of Polyphenol and Flavonoid Contents

The total polyphenol and flavonoid contents of the GBE were quantified using modified colorimetric methodologies. To assess total phenolic content, we employed a slightly altered Folin–Denis method, based on prior research [37]. In brief, we mixed 10 µL of a GBE sample (1 mg/mL) with 10 µL of 1 N Folin–Ciocalteu’s phenol reagent (Sigma-Aldrich, St. Louis, MO, USA) and 170 µL of distilled water, incubating the mixture in the dark at room temperature for 5 min. Then, we added 10 µL of a 20% Na_2_CO_3_ solution and let the mixture incubate in the dark for another 20 min at room temperature. Using a microplate reader (Molecular Devices, San Jose, CA, USA), we measured the absorbance at 765 nm. We obtained a calibration curve using gallic acid (Sigma-Aldrich), expressing the total phenolic content as milligrams of gallic acid equivalents per gram of dry extract (mg GAE/g). To quantify flavonoid content, we used the aluminum chloride colorimetric method, based on prior research, with minor modifications [38]. Briefly, we combined 10 µL of GBE sample (1 mg/mL), 10 µL of a 10% AlCl_3_ solution, 10 µL of 1 M potassium acetate, and 170 µL of distilled water. This mixture was incubated at room temperature for 30 min. We then measured the absorbance at 415 nm using a microplate reader and established a calibration curve using quercetin hydrate (TCI, Tokyo, Japan). The flavonoid content was expressed as milligrams of quercetin equivalents per gram of dry extract (mg QE/g).

### 2.4. Assessment of Lipid Peroxidation Levels

We evaluated lipid peroxidation by measuring the formation of malondialdehyde in brain tissue. In brief, we dissected the brain of a 7-week-old male ICR mouse immediately following intracardial perfusion. The brain tissue was homogenized in 10 volumes of cold phosphate-buffered saline (PBS) and centrifuged at 3000× *g* for 20 min at 4 °C to yield the supernatant. To stimulate lipid peroxidation, we combined 450 μL of the brain homogenate with 50 μM ferric chloride (25 μL), either in the presence or absence of pre- or post-fermented GBE (25 μL, 0–1000 μg/mL), and incubated the mixture for 30 min at 37 °C. We terminated the reaction by adding 500 μL of an 8.1% *w*/*v* sodium dodecyl sulfate solution and 1 mL of 20% acetic acid. After the supernatants cleared, we mixed 500 μL aliquots with an equal volume of a 0.8% *w*/*v* thiobarbituric acid solution and heated the Eppendorf tubes at 95 °C for 30 min. Following cooling on ice, we pipetted 100 μL of each sample into 96-well plates. We measured the absorbance at 532 nm using a microplate reader.

### 2.5. Evaluation of Antioxidant Scavenging Potential

The antioxidant scavenging potential of the pre- or post-fermented GBE was measured using the 1,1-diphenyl-2-picrylhydrazyl (DPPH) radical scavenging activity assay, following a method previously described [39]. Specifically, we added 10 μL of pre- or post-fermented GBE solution, at varying concentrations (0–1000 μg/mL) and dissolved in distilled water, to each well of a 96-well plate. Subsequently, we added 90 μL of a 0.2 mM DPPH solution, dissolved in methanol. Following a 10 min incubation period at room temperature, we measured the absorbance at 517 nm using a microplate reader. We calculated the DPPH radical scavenging activity using the following formula: % scavenging activity = (control absorbance − sample absorbance)/control absorbance × 100. We performed all sample analyses in triplicate.

### 2.6. Assessment of Hydroxyl Radical-Induced Oxidative Damage

The evaluation of hydroxyl radical-induced oxidative damage was executed through a metal-catalyzed reaction as described previously [40]. The target protein, bovine serum albumin (BSA), was prepared at a final concentration of 0.5 mg/mL in phosphate-buffered saline (PBS). This BSA solution was subjected to incubation both with and without 100 μM copper (Cu^2+^) and 2.5 mM hydrogen peroxide (H_2_O_2_), in the presence or absence of pre- or post-fermented GBE at varying concentrations (0–1000 μg/mL). The control antioxidant utilized was ascorbate at 100 μM, directly dissolved in PBS. All reaction mixtures were housed in open tubes and placed in a shaking water bath maintained at a constant temperature of 37 °C. Following the completion of reactions, we separated each mixture using 10% sodium dodecyl sulfate–polyacrylamide gel electrophoresis (SDS–PAGE). Subsequent to separation, we stained the gel with a 0.1% Coomassie blue solution.

### 2.7. Determination of AChE Activity

The AChE activities were evaluated utilizing a modified Ellman’s method, as previously described [41]. Briefly, the brain of an ICR mouse was dissected promptly following intracardial perfusion. The excised brain tissue was then homogenized in 20 volumes of cold phosphate-buffered saline (PBS), which contained a proteinase inhibitor cocktail (Sigma-Aldrich). For the measurement of AChE activity, the following components were mixed in a cuvette: 0.1 mL of the test sample, 2.5 mL of PBS, 0.1 mL of 10 mM 5,5’-dithio-bis-2-nitrobenzoic acid (Sigma-Aldrich), and 0.2 mL of brain homogenate. Subsequently, 0.1 mL of 75 mM acetylthiocholine iodide (Sigma-Aldrich) was added to the cuvette mixture. The enzymatic activity was recorded at 412 nm for a duration of 5 min, at 1 min intervals, using a microplate reader. The results were normalized and presented as a percentage of the control (vehicle) activity, using the formula: % activity = (sample activity/control activity) × 100. Each experiment was performed in triplicate.

### 2.8. Cell Culture and Maintenance

Human neural stem cells (F3) and choline acetyltransferase (ChAT)-overexpressing human neural stem cells (F3.ChAT) were cultured and maintained in a specialized medium. The culture medium consisted of Dulbecco’s modified Eagle’s medium (DMEM; Biowest, Nuaille’, Cholet, France) fortified with antibiotics (100 IU/mL penicillin and 100 µg/mL streptomycin). The medium was further supplemented with 10% heat-inactivated fetal bovine serum (Biowest). Cell cultures were incubated at 37 °C in an environment maintained at 5% CO_2_ and 95% air. For all experiments, cells were grown until they achieved a confluence of over 90% and were ensured to be within their first 20 passages to maintain optimal growth conditions and genetic stability.

### 2.9. Preparation and pH Adjustment of AF64A

AF64A was freshly prepared from acetyl-AF64A following a method described previously [40]. Acetyl-AF64A (Sigma-Aldrich) was rapidly dissolved in distilled water under constant stirring. The pH was initially adjusted to and maintained between 10.5 and 11.0 using a NaOH solution for a duration of 30 min. Following this, an HCl solution was added to decrease the pH to a range of 5.5–7.0. The final pH adjustment was made to 7.4 with the addition of a NaOH solution. The final concentration of AF64A was adjusted to 10.0 mM with the addition of distilled water. This stock solution was then further diluted with phosphate-buffered saline (PBS) to reach the desired working concentrations for the experiments.

### 2.10. Assessment of Cell Viability and Protective Effects of Pre- and Post-Fermented GBE

Cell viability following treatment with pre- or post-fermented GBE and AF64A was evaluated using the Cell Counting Kit-8 (CCK-8; Dojindo Laboratories, Japan). F3 and F3.ChAT cells were seeded at a density of 1 × 10^4^ cells/well in a 96-well plate. After treatment with varying concentrations of pre- or post-fermented GBE (0–1000 µg/mL) and AF64A (0–1000 µM), 10 µL of CCK-8 was added to each well (containing 100 µL of culture medium) and the plate was incubated for 24 h at 37 °C. The number of viable cells was determined by measuring the absorbance at 450 nm using a microplate reader. The protective effects of pre- and post-fermented GBE against AF64A-induced cytotoxicity were assessed by measuring the release of lactate dehydrogenase (LDH) from F3 and F3.ChAT cells. LDH content in the culture supernatants was determined using a commercial LDH assay kit (Promega, Madison, WI, USA), which is based on a coupled enzymatic reaction that converts a tetrazolium salt into a red formazan product. The intensity of the formazan color, proportional to the number of lysed cells, was quantified by measuring the absorbance at 490 nm using a microplate reader. All experiments were conducted in triplicate.

### 2.11. Induction of Cognitive Deficits and Treatment in Mice

Six-week-old male ICR mice (SPF, certificate #211005) were obtained from Daehan Biolink (Daehan Biolink, Eumseong, Republic of Korea). The mice were kept in an environment with a constant temperature (23 ± 3 °C) and relative humidity (50 ± 10%), following a 12 h light/dark cycle. Standard rodent diets and purified water were available ad libitum. Following 1 week of acclimation to the laboratory conditions, the mice were randomly assigned to one of the following groups: Normal Control group (NC; *n* = 7), AF64A + Vehicle group (Veh; *n* = 7), AF64A + Donepezil group (Donepezil; *n* = 7), and AF64A + post-fermented GBE at 100 mg/kg (GBE100; *n* = 7), 200 mg/kg (GBE200; *n* = 7), and 400 mg/kg (GBE400; *n* = 7). Mice were anesthetized using ketamine (Yuhan Corporation, Seoul Republic of Korea) and rompun (Bayer Korea, Seoul, Republic of Korea) and positioned in a stereotaxic frame. To induce cognitive deficits, AF64A cholinotoxin was administered as previously described [18]. Following the skin incision, a fresh AF64A solution (3 nmol/3 μL/mouse) was infused into the right ventricle at the following stereotaxic coordinates from the bregma: posterior 0.6 mm, lateral 1.1 mm, and ventral 2.0 mm, at a flow rate of 0.5 μL/min. After a 2-week recovery period, either vehicle (0), 100, 200, and 400 mg/kg of GBE, or 2 mg/kg of donepezil, was orally administered once a day for 6 weeks (Figure 1). Prior to administration, mice were fasted for 4 h, with food and water provided ad libitum post-administration. Vehicle administration was performed concurrently with other study compounds for comparison purposes. All animal experiments were conducted following the Standard Operation Procedures of the Laboratory Animal Center at Chungbuk National University (CBNU, MFDS Approval #2), Republic of Korea. The protocol was approved by the Institutional Animal Care and Use Committee of CBNU (#CBNUA-1398-20-01).

### 2.12. Assessment of Passive Avoidance Memory Performance

The passive avoidance memory performance of mice was evaluated using a step-through Shuttle box (ENV-010MD; Med Associates, St. Albans, VT, USA). This evaluation was carried out daily for seven days, beginning 8 weeks after the AF64A injection. The Shuttle box apparatus comprises two compartments, one illuminated and the other dark. Both compartments are equipped with a steel-grid floor for the delivery of an electric shock. During the trials, mice were placed in the illuminated compartment, and the door to the dark compartment was opened. As the mice ventured into the dark compartment, they received an electric shock (0.4 mA for 2 s). After that, the latency time, defined as the time it took the mice to reenter the dark compartment after the light in the illuminated compartment was turned on, was recorded. A maximum latency time of 180 s was set as the cutoff for successful memory acquisition.

### 2.13. Assessment of Spatial Memory Using Morris Water Maze

The evaluation of spatial memory was conducted using a Morris water maze system (Smart v2.5; Panlab Technology, Barcelona, Spain), which was connected to a closed-circuit television (CCTV) system (Samsung, Changwon, Republic of Korea). The testing schedule aligned with the passive avoidance performance assessment and occurred 2 h later on the same day. The water maze trials utilized a circular water bath of 150 cm in diameter, filled with water to a depth of 27 cm. The water temperature was consistently maintained at 22 ± 2 °C. The bath was divided into four quadrants, with a hidden escape platform (10 cm in diameter, 25 cm in height) situated in the center of one of the quadrants. This platform was submerged 2 cm beneath the water’s surface and was concealed using squid ink. Throughout the trials, the mice were trained to locate the concealed platform using various external cues. The positions of these cues remained constant for the duration of the experiments. The escape latency, defined as the time it took for the mice to find the platform during the trials, was recorded by tracking their movements via the CCTV monitor.

### 2.14. Quantitative Real-Time Polymerase Chain Reaction (PCR) Analysis

Total RNA was extracted from F3, F3.ChAT cells, and brain tissues using TRIzol Reagent (Invitrogen, Carlsbad, CA, USA), following the manufacturer’s instructions. The real-time quantitative PCR was performed as described in prior literature. The housekeeping gene, glyceraldehyde 3-phosphate dehydrogenase (GAPDH), served as an internal standard to normalize the expression of target transcripts. Primers used to amplify the transcripts of choline transporter (CHT), choline acetyltransferase (ChAT), vesicular acetylcholine transporter (VAChT), nicotinic acetylcholine receptor (nAChR) α5 and β2, muscarinic1 acetylcholine receptor (m1AChR), nerve growth factor (NGF), and brain-derived growth factor (BDNF) are listed in Appendix A. Data from three independent assays, each conducted in triplicate, were analyzed using the comparative Ct method [42].

### 2.15. Western Blot Analysis

Both the F3 and F3.ChAT cells, as well as brain tissues, were homogenized in 10 volumes of Radioimmunoprecipitation Assay (RIPA) buffer (Thermo Fisher Scientific, Waltham, MA, USA), supplemented with protease (Sigma-Aldrich) and phosphatase (Sigma-Aldrich) inhibitors. The Western blotting procedure was performed as previously described [12]. Membranes were subjected to immunoblotting with the appropriate primary antibodies, which was followed by incubation with horseradish peroxidase-conjugated anti-rabbit and anti-mouse secondary antibodies (Jackson ImmunoResearch Laboratories, Inc., West Grove, PA, USA). The specific antibodies utilized in this study are detailed in Appendix A. Band densities were quantified using ImageJ software (NIH, Bethesda, MD, USA) and were normalized to the actin controls.

### 2.16. Glial Fibrillary Acidic Protein (GFAP) Immunostaining of Brain Tissue

The brain tissues of the mice were perfused with a 10% paraformaldehyde solution, followed by a 48 h post-fixation in the same solution. Tissues were then cryoprotected in a 30% sucrose solution for 72 h. Coronal cryosections, 1.0 mm posterior to the bregma and 30 μm thick, were prepared. For the immunohistochemical staining of astrocytic GFAP, brain cryosections were rinsed in Tris-buffered saline (TBS) and treated with 3% hydrogen peroxide for 5 min to block endogenous peroxidase activity. After rinsing with TBS and a blocking step with 5% BSA, the sections were incubated at 4 °C overnight with a GFAP-specific antibody (1:300; rabbit polyclonal, Chemicon, Temecula, CA, USA). This was followed by incubation with a secondary antibody conjugated with Alexa Fluor-594 (1:300; Molecular Probes, Eugene, OR, USA). The sections were counterstained with 4′,6-diamidino-2-phenylindole (DAPI, Sigma-Aldrich) to visualize cellular nuclei. GFAP staining was used in combination with DAPI staining to identify astrocytes based on their characteristic morphology and location within the brain tissue. Following staining, all samples were immediately evaluated and photographed using a fluorescence microscope (EVOS FL Auto2 Cell imaging system; Thermo Fisher Scientific, Waltham, MA, USA). The area of GFAP-positive cells was quantified using ImageJ software.

### 2.17. Brain Acetylcholine Concentration Analysis

To measure the concentration of acetylcholine, the mice’s brains were weighed and homogenized in nine volumes of phosphate-buffered saline (PBS) that included a protease inhibitor cocktail (Sigma-Aldrich). The protease inhibitors were included to prevent the degradation of acetylcholine by acetylcholinesterase, ensuring an accurate measurement of its concentration. The brain tissue was rapidly collected and processed after the mice were sacrificed to reduce ischemic damage and inhibit post-mortem metabolism. After centrifugation at 13,500× *g* for 6 min at 4 °C, the supernatant’s acetylcholine concentration was measured using the Amplex Red acetylcholine/acetylcholinesterase assay kit (Molecular Probes, Eugene, OR, USA) according to the manufacturer’s instructions. In this assay, acetylcholinesterase hydrolyzes acetylcholine to produce choline, which is then oxidized into betaine and H_2_O_2_ by choline oxidase. H_2_O_2_ reacts with Amplex Red (7-dihydroxyphenoxazine) in the presence of horseradish peroxidase, generating the highly fluorescent compound, resorufin. The resulting fluorescence was measured using a fluorescence microplate reader (Molecular Devices) with an excitation range of 530–560 nm and emission detection at 590 nm. It is important to note that the control group’s reported value of almost 1 mmol/g of tissue may vary, so it should be interpreted with caution.

### 2.18. Statistical Analyses

Statistical analyses were carried out using a one-way ANOVA, followed by Tukey’s post hoc test. The normality of the data was not explicitly tested prior to applying ANOVA, but the assumption of normality was inferred through the visual inspection of histograms and normal probability plots where necessary. All analyses were carried out using the Statistical Package for the Social Sciences for Windows software (version 12.0; SPSS Inc., Chicago, IL, USA). Statistical significance was set at *p* < 0.05. All data are presented as the mean ± standard deviation (SD).

## 3. Results

### 3.1. Composition of Ginsenosides, Polyphenols, and Flavonoids in Pre- and Post-Fermented GBE

To investigate the effect of GBE fermentation, we analyzed the contents of pre- and post-fermented GBE using HPLC. As shown in Figure 2 and Appendix A, the pre-fermented GBE was primarily composed of Re, Rd, Rb1, Rb2, and Rc, while the post-fermented GBE exhibited significant increases in Rg3(S), Rg3(R), Rk1, and Rg5 compared to the pre-fermented GBE. Interestingly, the contents of total polyphenols and flavonoids also significantly increased after fermentation, from 24.26 ± 0.23 to 28.40 ± 0.10 mg GAE/g and from 3.95 ± 0.07 to 4.71 ± 0.12 mg QE/g, respectively.

### 3.2. Comparative Physiological Properties of Pre- and Post-Fermented GBE

To investigate the physiological properties of pre- and post-fermented GBE, we analyzed their effects on TBARS, DPPH, AChE inhibitory effect, and hydroxyl radical-mediated oxidation using assays. As shown in Figure 3A, treatment with FeCl_3_ (50 μM) resulted in a 2.4-fold increase in MDA production, which was reduced by both pre- and post-fermented GBE. The antioxidative effects of post-fermented GBE were significantly different at 125 μg/mL compared to pre-fermented GBE. Similarly, the radical scavenging activity (RSA) of post-fermented GBE was significantly increased at 250 μg/mL, while the RSA of pre-fermented GBE was only significant at 500 μg/mL (Figure 3B). In addition, the activity of AChE decreased with increasing concentrations of pre- and post-fermented GBE, significantly reducing from 16 μg/mL in pre-fermented GBE to 2 μg/mL in post-fermented GBE (Figure 3C). The inhibitory effect of AChE activity in post-fermented GBE was significantly different at 250 μg/mL compared to pre-fermented GBE. Both pre- and post-fermented GBE exhibited antioxidative effects in the hydroxyl radical-mediated oxidation assay at 125 μg/mL, with the effect of post-fermented GBE being generally superior to that of pre-fermented GBE (Figure 3D). Subsequent experiments were therefore conducted using post-fermented GBE.

### 3.3. Protective Effects of Post-Fermented GBE on AF64A-Induced Damage in F3 and F3.ChAT Cells

To confirm the protective effect, the cytotoxicity of post-fermented GBE and AF64A was evaluated at various concentrations (0–1000 μM, respectively) on F3 and F3.ChAT cells. Overall, post-fermented GBE was almost non-toxic, but toxicity was observed at the highest concentration (1000 μg/mL) (Figure 4A). In the case of AF64A (Figure 4B), toxicity was observed at 63 μM, and the experiment was conducted at 100 μM, at which 80% of the F3 and F3.ChAT cells survived. The LDH assay showed that AF64A significantly increased the release of LDH from the F3 and F3.ChAT cells (Figure 4C,D); however, post-fermented GBE treatment significantly decreased it in a dose-dependent manner. However, treatment with donepezil (1 μM) as a reference drug did not show a protective effect.

### 3.4. Post-Fermented GBE’s Effects on Cholinergic Pathway in AF64A-Damaged F3 and F3.ChAT Cells

To investigate the neuroprotective effects of post-fermented GBE, the genes related to the cholinergic pathway in F3 (Figure 5) and F3.ChAT (Figure 6) cells were analyzed by real-time PCR. As expected, AF64A significantly reduced the gene expression related to acetylcholine synthesis, including CHT, ChAT, and VAChT, as well as the cholinergic receptors nAChR α5, nAChR β2, and m1AChR, and the growth factors NGF and BDNF. However, treatment with post-fermented GBE significantly increased the expression of these genes in a concentration-dependent manner. Coincidently, Western blot analysis showed that AF64A treatment (100 μM) decreased the expression of BDNF, NGF, ChAT, and VAChT, while treatment with post-fermented GBE increased their expression (Figure 7). Treatment with donepezil (1 μM) also increased the expression of these genes and proteins; however, the effect of post-fermented GBE was overall better than that of donepezil.

### 3.5. Restoration of Cognitive Function in AF64A-Induced Memory Deficit Animals

Two weeks after AF64A injection, memory deficit was confirmed in some animals using passive avoidance and Morris water maze performances. The mice were then orally administered with post-fermented GBE (0, 100, 200, and 400 mg/kg) and donepezil (2 mg/kg) for 6 weeks, during which no adverse effects such as weight loss were observed (Appendix A). If no significant changes in body weight were observed, we proceeded with the next phase of cognitive testing. After 6 weeks, learning and memory function in mice were analyzed by passive avoidance and Morris water maze performances (Figure 8). In the passive avoidance performance, the retention time of animals in the normal group increased from the 3rd trial, reaching a maximum from the 5th trial. The retention time of animals in the AF64A + vehicle group did not increase. However, the retention time of animals in the AF64A + post-fermented GBE group increased, especially at 400 mg/kg, with the time increasing from the 4th trial and reaching a maximum from the 5th trial, similar to the normal group. Similarly, in the Morris water maze performance, the escape time of animals in the normal group decreased as the number of trials increased; however, the AF64A + vehicle group did not decrease. However, animals in the AF64A + post-fermented GBE group had a reduced escape time as in the normal group.

### 3.6. Cholinergic Pathway Modulation by Post-Fermented GBE in AF64A-Induced Brain Damage in Mice

The effect on the cholinergic pathway was analyzed by real-time RT-PCR in cells (Figure 9). In the AF64A + vehicle group, there was a significant reduction in the gene expression of CHT, ChAT, VAChT, nAChR α5, nAChR β2, m1AChR, NGF, and BDNF. However, animals treated with post-fermented GBE in the AF64A + group showed a dose-dependent increase in the expression of these genes.

In the Western blotting (Figure 10), the expression of CHT, ChAT, VAChT, BDNF, and NGF was significantly decreased by AF64A injection. However, their expression was increased by the administration of post-fermented GBE. Also, the expression of GFAP and amyloid beta (Aβ) was significantly increased by AF64A injection. However, their expression was significantly decreased by the administration of post-fermented GBE.

### 3.7. GFAP Inactivation by Post-Fermented GBE in the Context of AF64A-Induced Brain Damage in Mice

Inflammation-activated central nervous system astrocytes are characterized by hypertrophy and proliferation, along with an upregulation of their cytoskeletal protein, GFAP. As shown in Figure 11, intensive GFAP-immunoreactivities (red-colored) were observed in activated astrocytes in the subventricular zone of AF64A-injected mice. However, astrocytic activation was significantly inhibited by the administration of post-fermented GBE (100–400 mg/kg) to normal levels in the subventricular zone.

### 3.8. ACh Concentration Improvement in AF64A-Induced Brain Damage through Post-Fermented GBE Administration

Our study revealed a significant increase in the number of activated astrocytes in the vehicle group following AF64A injection. However, the administration of post-fermented GBE reduced the number of these activated astrocytes in a dose-dependent manner, as shown in Figure 12A, where the count of activated astrocytes was measured using ImageJ software. Additionally, in the vehicle group, we observed a significant reduction in ACh levels after the AF64A injection, as depicted in Figure 12B. Intriguingly, post-fermented GBE administration led to a dose-dependent recovery of ACh concentration. Notably, we observed a direct correlation between the increase in ACh levels and the dosage of post-fermented GBE administered.

## 4. Discussion and Conclusions

The present study investigated the potential cognitive benefits of fermented GBE in a rodent model of AF64A-induced memory deficits. In this study, the active ingredients of GBE were found to be enhanced after fermentation, as evidenced by a chromatographic analysis comparing pre- and post-fermented GBE. Furthermore, the treatment with fermented GBE restored the cholinergic nervous system in human neural stem cells and animals damaged by AF64A. Thus, these findings suggest that fermented GBE could hold the potential for treating memory deficits in clinical applications. Ginseng has long been utilized in traditional oriental medicine due to its therapeutic effects, which are primarily attributed to its bioactive components, ginsenosides, that are extensively studied in ginseng roots [43]. Recently, ginseng berries have been considered a potential alternative source of bioactive compounds, as they contain a relatively high concentration of ginsenosides, primarily Re and Rd [23]. Additionally, the bioactivity of ginseng can be enhanced through processing methods, such as red ginseng preparation, which can result in higher levels of ginsenosides [44,45]. Our previous studies demonstrated that fermenting ginseng berries using *L. plantarum* can alter the composition of ginsenosides and increase their pharmacological activities [35,46]. Similarly, in this study, we confirmed that the content of Rg3, total polyphenols, and flavonoids increased after fermentation with *L. plantarum*.

The change in ginsenoside composition led to altered physiological activity, resulting in increased antioxidant and anti-inflammatory effects, as confirmed by the DPPH assay, hydroxyl radiated oxidation assay, and MDA assay. Additionally, the AChE activity test showed an inhibitory effect on AChE activity in both pre- and post-fermented GBE. The key ginsenosides of pre-fermented GBE, Re and Rc, are known to regulate the cholinergic nervous system [47,48], while Rg3, Rg5, and Rk1 in post-fermented GBE are reported to inhibit AChE activity [48,49]. In vitro and in vivo tests were conducted to investigate the effects of post-fermented GBE on the cholinergic nervous system. AF64A, a cholinergic toxin, is widely used in screening drugs for the development of dementia treatments [41,50]. In this study, AF64A induced overall inhibition of the cholinergic nervous system in human neural stem cells, reduced the concentration of acetylcholine in brain tissue, and increased the expression of GFAP, an indicator of inflammation and brain damage in animal tests [12,42,51,52]. However, treatment with post-fermented GBE counteracted the effect of AF64A, enhancing the activity of the cholinergic nervous system. This increase in activity was attributed to increased expression of NGF and BDNF, which prevent nerve cell death and promote their proliferation, as well as increased activity of the cholinergic nervous system [12,53,54,55]. Thus, an increase in ginsenosides, such as Rg3, Rg5, and Rk1, in post-fermented GBE seems to inhibit inflammation caused by AF64A and increase antioxidant effects to protect cells [56,57], as well as increase BDNF and NGF to regulate the expression of the cholinergic nervous system [58,59,60]. Interestingly, the administration of AF64A led to an increase in amyloid-beta, which is thought to be due to the change of the amyloidogenic pathway by increased inflammation [41]. Additionally, the administration of GBE reduced the level of Aβ. Although this study demonstrated that the cholinergic nervous system was restored by increasing the levels of BDNF and NGF through the administration of GBE, further research is needed to investigate the reduction of Aβ through the administration of post-fermented GBE. These findings strongly suggest that the robust antioxidant properties of fermented GBE could aid in protecting against oxidative stress, known to contribute to cognitive decline. Furthermore, it may modulate neurotransmitter systems, such as the cholinergic system, as well as Aβ levels, which play crucial roles in memory, learning, and neuropathogenesis.

Although our study provides valuable insights into the effects of pre- and post-fermented GBE and AF64A on cholinergic neurons, it is important to recognize the limitations of our experimental model. Despite these, the findings from this research carry significant clinical implications for the development of new treatments for Alzheimer’s disease and other cognitive disorders. Future studies employing more complex and comprehensive models of AD could provide further insights into the effects of these treatments on AD pathogenesis.

In summary, this study demonstrates that GBE fermented by *L. plantarum* leads to an increase in bioactive compounds, such as Rg3, Rg5, Rk1, total polyphenols, and flavonoids, compared to pre-fermented GBE. This results in enhanced pharmacological activities, including antioxidant and anti-inflammatory effects. Notably, post-fermented GBE also seems to upregulate the expression of NGF and BDNF while reducing AChE activity. These effects may help protect against brain damage and restore memory in AF64A-induced memory-deficient animals. Overall, these findings suggest that fermented GBE holds potential as a therapeutic agent for AD and other neurodegenerative disorders. However, further research is needed to elucidate the underlying mechanisms of the observed effects and to determine the optimal dosage and treatment duration for maximum efficacy. Additionally, future studies should employ more comprehensive and accurate models of AD that better represent the disease’s pathological features to assess the therapeutic potential of fermented GBE more reliably. This would aid in determining the feasibility of fermented GBE as a treatment for AD and related neurodegenerative disorders. Moreover, clinical trials involving human subjects are necessary to evaluate the safety, tolerability, and efficacy of fermented GBE in real-world settings. By conducting these future studies, researchers can build upon the current findings and potentially develop novel treatment options for patients suffering from Alzheimer’s disease and other neurodegenerative conditions.

AD is a progressive neurodegenerative disorder that currently has no known curative treatment. However, there are several therapeutic and preventive treatments available that can help to manage symptoms and slow disease progression. These treatments include medications like AChE inhibitors (donepezil, rivastigmine, and galantamine) and memantine, as well as non-pharmacological interventions like cognitive stimulation therapy and physical exercise. While these treatments can provide some relief to patients, there is still a need for more effective treatments and preventative measures due to an incomplete understanding of the pathophysiology of AD. Therefore, there is a growing interest in the application of natural extracts as potential therapeutic agents for AD. Our study provides valuable insights into the potential cognitive benefits of fermented GBE, underscoring the need for continued research to fully comprehend its clinical applications. The findings suggest that fermented GBE, rich in the major active ginsenosides verified in this study, holds promise as a candidate for developing new therapeutic interventions for memory deficits and cognitive disorders. However, it is important to note that the current study was conducted in animal models, and further research is needed to determine the safety, tolerability, and efficacy of fermented GBE in human populations. Future clinical trials involving human subjects will be necessary to evaluate the potential therapeutic benefits of fermented GBE in real-world settings. By conducting these future studies, researchers can build upon the current findings and potentially develop novel treatment options for patients suffering from Alzheimer’s disease and other neurodegenerative conditions.

In conclusion, our study sheds light on the potential cognitive benefits of fermented GBE, thus emphasizing the necessity for further research to fully unravel its clinical implications. The results imply that fermented GBE, enriched with the primary active ginsenosides confirmed in our study, presents itself as a promising avenue for the creation of novel therapeutic interventions to address memory impairments and cognitive dysfunctions. Nevertheless, it is crucial to recognize that our study relied on animal models, warranting subsequent studies to validate the safety, tolerability, and effectiveness of fermented GBE in human subjects. Future clinical trials involving human participants will be indispensable for gauging the potential therapeutic advantages of fermented GBE in practical scenarios. By executing these prospective investigations, scientists can expand on our current knowledge and potentially formulate innovative treatment strategies for patients afflicted with Alzheimer’s disease and other neurodegenerative disorders.

## Figures and Tables

**Figure 1 nutrients-15-03389-f001:**
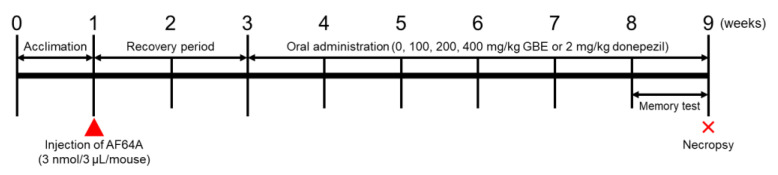
Schematic illustration of the experimental design. Six-week-old male ICR mice were randomly allocated into different groups: normal control group (NC; *n* = 7), AF64A + vehicle group (Veh; *n* = 7), AF64A + Donepezil group (Donepezil; *n* = 7), AF64A + post-fermented GBE 100 mg/kg group (GBE100; *n* = 7), AF64A + post-fermented GBE 200 mg/kg group (GBE200; *n* = 7), and AF64A + post-fermented GBE 400 mg/kg group (GBE400; *n* = 7). Two weeks following AF64A injection (3 nmol/3 μL/mouse), the mice were treated with GBE (100, 200, and 400 mg/kg) or donepezil (2 mg/kg) over a duration of 9 weeks. The doses of GBE and donepezil were administered orally and their frequencies varied as indicated by the arrows. Cognitive performance was evaluated at the beginning and during the 3rd and 9th weeks. The Morris water maze test was employed to evaluate spatial learning and memory, while the novel object recognition test was used to assess recognition memory.

**Figure 2 nutrients-15-03389-f002:**
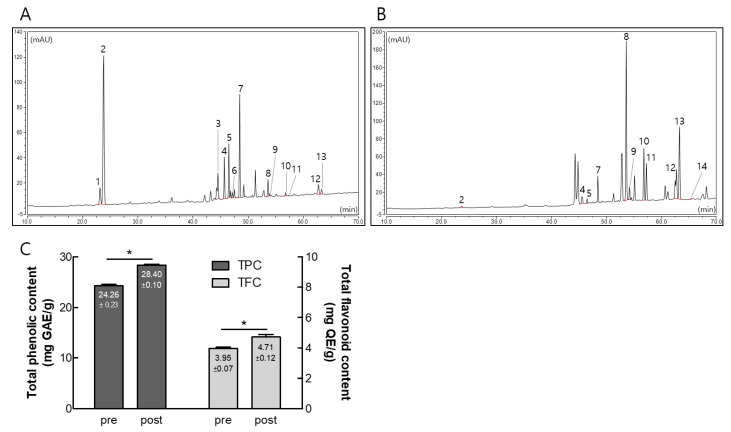
Contents of ginsenoside, total polyphenols, and flavonoids in pre- and post-fermented ginseng berry extract (GBE). (**A**,**B**) High-performance liquid chromatography (HPLC) chromatograms of ginsenosides in pre- (**A**) and post- (**B**) fermented GBE. Peaks were analyzed based on HPLC chromatograms of the ginsenoside standard mixture. (**C**) Total amounts of phenol (mg gallic acid equivalents per gram of dry extract) and flavonoid (mg quercetin equivalents per gram of dry extract) compounds. Data are presented as mean ± SD in triplicates. * *p* < 0.05, significantly different from pre-fermented GBE.

**Figure 3 nutrients-15-03389-f003:**
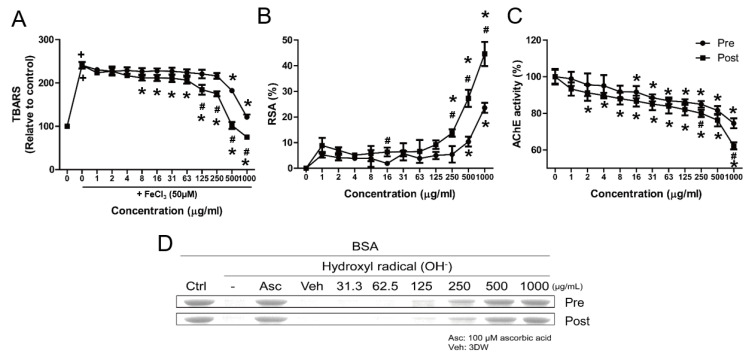
Physiological properties of pre- and post-fermented GBE. (**A**) Inhibitory activity on FeCl_3_-induced lipid peroxidation (thiobarbituric acid reactive substances (TBARS) production), (**B**) 1,1-diphenyl-2-picrylhydrazyl (DPPH) radical-scavenging activity (RSA) in 10 min, (**C**) acetylcholinesterase (AChE) activity in brain tissue treated with pre- and post-fermented GBE, and (**D**) hydroxyl radical-mediated oxidation assay using H_2_O_2_ and BSA. All samples were analyzed in triplicate. Data are presented as mean ± SD in triplicates. + *p* < 0.05, significantly different from normal control; * *p* < 0.05, significantly different from vehicle; # *p* < 0.05, significantly different from pre-fermented GBE.

**Figure 4 nutrients-15-03389-f004:**
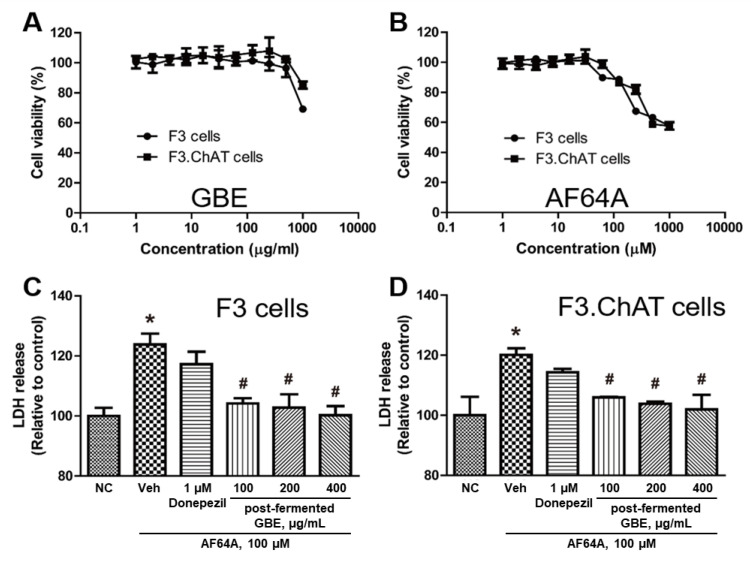
Cytotoxicity and protective effect of post-fermented GBE in F3 and F3.ChAT cells. (**A**,**B**) Cell viability of F3 and F3.ChAT cells was treated with various doses of post-fermented GBE (0 to 1000 μg/mL) (**A**) and various doses of AF64A (0 to 1000 μM) (**B**). (**C**,**D**) Protective effect of post-fermented GBE on AF64A-induced damage in F3 cells (**C**) and F3.ChAT cells. Donepezil was used as a reference control. LDH, lactate dehydrogenase. Data are presented as mean ± SD in triplicates. * *p* < 0.05, significantly different from normal control (NC); # *p* < 0.05, significantly different from vehicle.

**Figure 5 nutrients-15-03389-f005:**
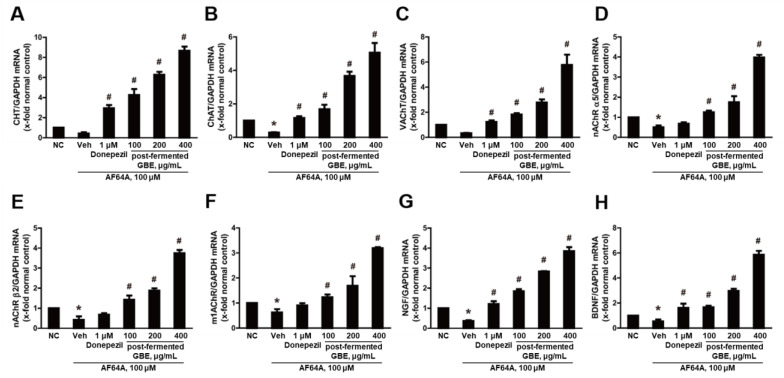
Effects of post-fermented GBE on the cholinergic pathway in AF64A-induced damage in human neural stem cells (F3 cells). (**A**) Choline transporter (CHT), (**B**) choline acetyltransferase (ChAT), (**C**) vesicular acetylcholine transporter (VAChT), (**D**) nicotinic acetylcholine receptors (nAChR) α5, (**E**) nAChR β2, (**F**) muscarinic 1 acetylcholine receptor (m1AChR), (**G**) nerve growth factor (NGF), and (**H**) brain-derived neurotrophic factor (BDNF) gene expression were analyzed by real-time PCR and normalized to GAPDH. Data are presented as mean ± SD in triplicates. * *p* < 0.05, significantly different from normal control (NC); # *p* < 0.05, significantly different from vehicle.

**Figure 6 nutrients-15-03389-f006:**
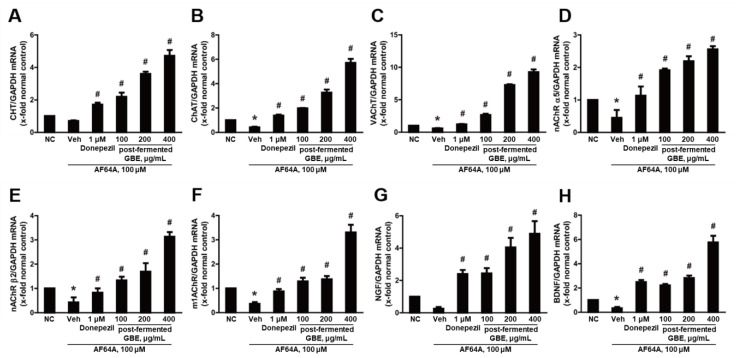
Effects of post-fermented GBE on the cholinergic pathway in AF64A-induced damage in ChAT overexpressing human neural stem cells (F3.ChAT cells). (**A**) CHT, (**B**) ChAT, (**C**) VAChT, (**D**) nAChR α5, (**E**) nAChR β2, (**F**) m1AChR, (**G**) NGF, and (**H**) BDNF gene expression were analyzed by real-time PCR and normalized to GAPDH. Data are presented as mean ± SD in triplicates. * *p* < 0.05, significantly different from normal control (NC); # *p* < 0.05, significantly different from vehicle.

**Figure 7 nutrients-15-03389-f007:**
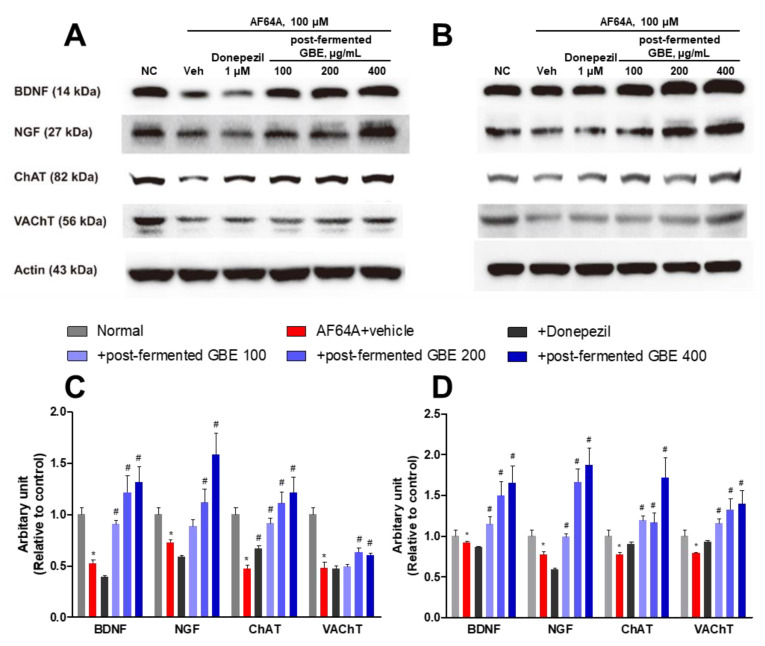
Effects of post-fermented GBE on protein expression in AF64A-induced damage in F3 (**A**,**C**) and F3.ChAT cells (**B**,**D**). (**A**,**B**) Representative bands of F3 (**A**) and F3.ChAT (**B**) cells. Protein expression was analyzed by Western blotting. (**C**,**D**) The band densities of F3 (**C**) and F3.ChAT (**D**) cells were quantified in arbitrary units and normalized to actin. Data are presented in triplicates as mean ± SD. * *p* < 0.05, significantly different from normal control (NC); # *p* < 0.05, significantly different from vehicle.

**Figure 8 nutrients-15-03389-f008:**
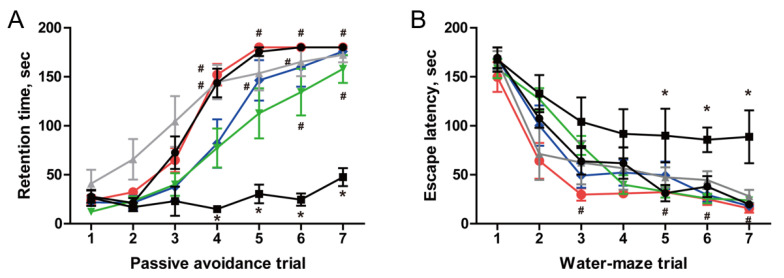
Recovery of cognitive function in AF64A-induced memory deficit animals (*n* = 7 in each group): (**A**) passive avoidance performance and (**B**) Morris water maze performance. The endpoint was set at 180 s if the animals stayed in the light chamber in the passive avoidance test or failed to find the platform in the Morris water maze test. ●, Normal; ■, AF64A (3 nmol) + vehicle; ▲, AF64A (3 nmol) + Donepezil (2 mg/kg); ▼, AF64A (3 nmol) + post-fermented GBE (100 mg/kg); ◆, AF64A (3 nmol) + post-fermented GBE (200 mg/kg); ●, AF64A (3 nmol) + post-fermented GBE (400 mg/kg). Data are presented as mean ± SD in triplicates. * *p* < 0.05, significantly different from normal control (NC); # *p* < 0.05, significantly different from vehicle.

**Figure 9 nutrients-15-03389-f009:**
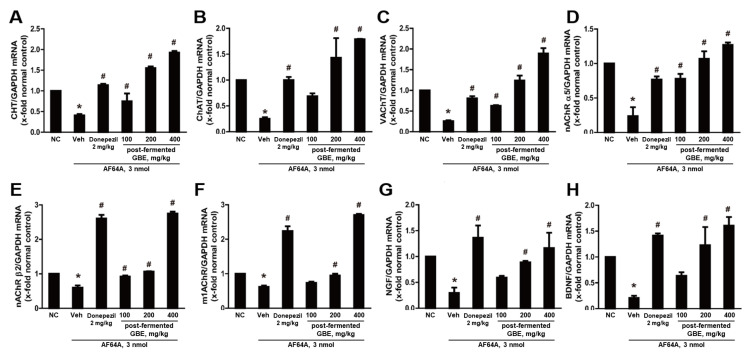
Effects of post-fermented GBE on the cholinergic pathway in AF64A-induced brain damage in mice (n = 5 in each group). (**A**) CHT, (**B**) ChAT, (**C**) VAChT, (**D**) nAChR α5, (**E**) nAChR β2, (**F**) m1AChR, (**G**) NGF, and (**H**) BDNF gene expression were analyzed by real-time PCR and normalized to GAPDH. Data are presented as mean ± SD in triplicates. * *p* < 0.05, significantly different from normal control (NC); # *p* < 0.05, significantly different from vehicle.

**Figure 10 nutrients-15-03389-f010:**
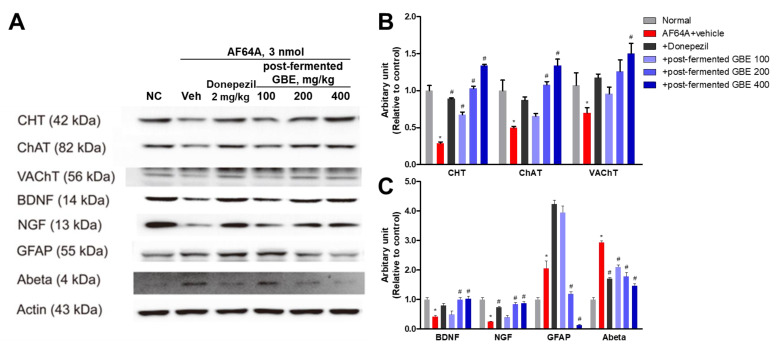
Effects of post-fermented GBE on protein expression in AF64A-induced brain damage in mice (*n* = 5 in each group). (**A**) Representative bands of brain tissues in mice. Protein expression was analyzed by Western blotting. (**B**,**C**) The band densities of brain tissues in mice were quantified in arbitrary units and normalized to actin. Data are presented as mean ± SD in triplicates. * *p* < 0.05, significantly different from normal control (NC); # *p* < 0.05, significantly different from vehicle.

**Figure 11 nutrients-15-03389-f011:**
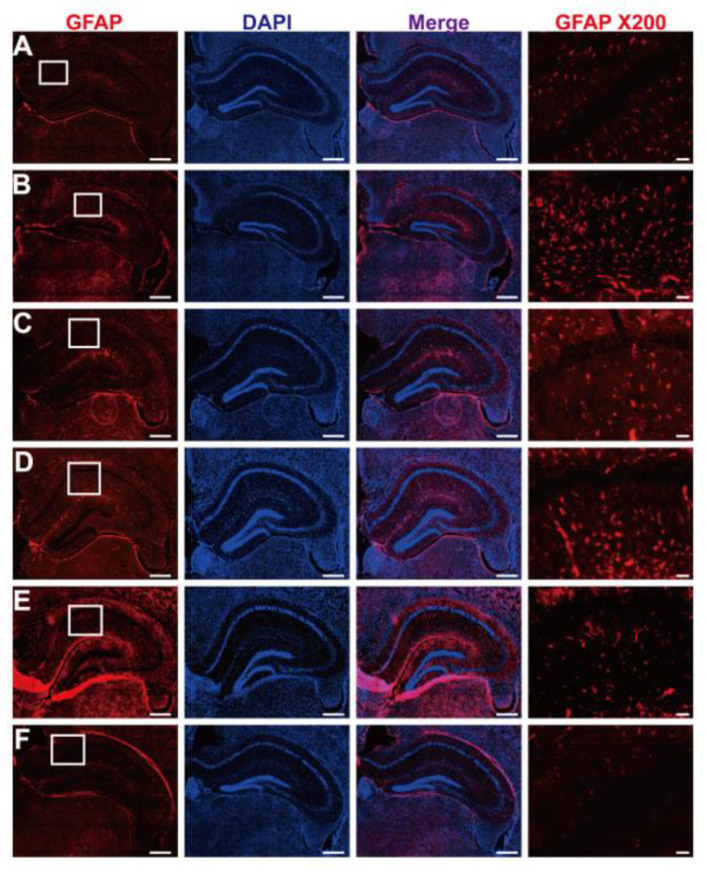
Inactivation of Glial Fibrillary Acidic Protein (GFAP) by post-fermented GBE in AF64A-induced brain damage in mice (*n* = 2 in each group). (**A**–**F**) These panels present representative microscopic images of activated (GFAP-positive) astrocytes, colored in red, scale bar = 500 μm. DAPI was used as a counterstain. The white square in the GFAPx200 image shows an enlarged photograph of GFAP, scale bar = 50 μm. (**A**) Normal, (**B**) AF64A (3 nmol) + vehicle, (**C**) AF64A (3 nmol) + Donepezil (2 mg/kg), (**D**) AF64A (3 nmol) + post-fermented GBE (100 mg/kg), (**E**) AF64A (3 nmol) + post-fermented GBE (200 mg/kg), and (**F**) AF64A (3 nmol) + post-fermented GBE (400 mg/kg).

**Figure 12 nutrients-15-03389-f012:**
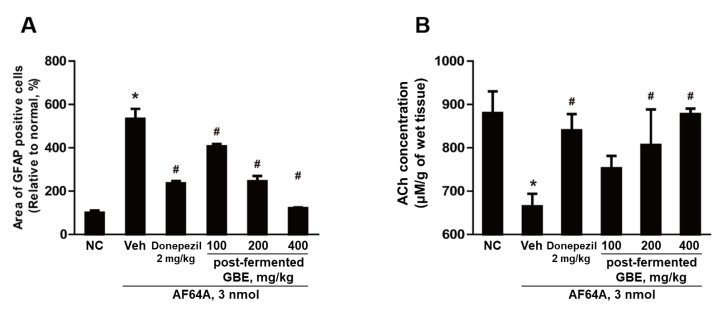
Illustration of the effects of post-fermented GBE on activated astrocytes count and acetylcholine (ACh) concentration following AF64A-induced brain damage. (**A**) Quantification of activated astrocytes (GFAP-positive) was performed using ImageJ software (*n* = 2 in each group). (**B**) The concentration of ACh in brain tissue was evaluated using an acetylcholine/acetylcholinesterase assay kit, following the protocol outlined in the Materials and Methods section (*n* = 5 in each group). Data are presented as mean ± SD in triplicates. * *p* < 0.05, significantly different from normal control (NC); # *p* < 0.05, significantly different from vehicle.

## Data Availability

All data generated from this study are contained within the article.

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
