# Peer review of "Improvement of Cognitive Function by Fermented Panax ginseng C.A. Meyer Berries Extracts in an AF64A-Induced Memory Deficit Model"

_nutrients, 2023, doi:10.3390/nu15153389_

Round 1

Reviewer 1 Report

I reviewed the manuscript entitled Improvement of Cognitive Function by Fermented Panax ginseng C.A. Meyer Berries Extracts in an AF64A-induced Memory Deficit Model.

The author conducted in-depth research on Alzheimer's disease. I agree to accept this manuscript, but there are a few issues that need to be revised before proceeding.

1) When it comes to statistics, p should be italicized, but the author did not italicize all the p, such as Figure 3. Data are presented as mean ? SD. +p < 0.05 The author is requested to check and revise the issue in the full text.

2) About the titles of the references. In some references, all notional words are capitalized, while in others, only the first word is capitalized. Please write in a unified way. Some journals' names are abbreviations and some are full names. Please unify them. In vivo and in vitro should be italicized.

minor revision

Author Response

Thank you for your kind advise. I tried to my best for correcting according to your comments. 

1) When it comes to statistics, p should be italicized, but the author did not italicize all the p, such as Figure 3. Data are presented as mean. SD. +p < 0.05 The author is requested to check and revise the issue in the full text.

 - All texts were checked and revised accordingly. “p” was italicized throughout the text (page 8; line 352 and 370).

2) About the titles of the references. In some references, all notional words are capitalized; in others, only the first word is capitalized. Please write in a unified way. Some journals' names are abbreviations, and some are full names. Please unify them. In vivo and in vitro should be italicized.

 - All texts were checked and revised accordingly. References were revised and rewritten, and the format was standardized.

Reviewer 2 Report

In this paper, the ginsenoside content and physiological characteristics of fermented ginseng berry extract ( GBE ) were studied. Lactobacillus plantarum was used to ferment fresh ginseng C.A. Meyer berries. The experimental results showed that the contents of Rg3 ( S ), Rg3 ( R ), Rk1, Rg5, total polyphenols and flavonoids in fermented GBE were significantly increased. It can also inhibit the cytotoxicity induced by af64a and up-regulate the expression of CHT, ChAT, VAChT, nAChRα5 and β2, m1AChR, NGF and BDNF genes. The conclusion indicates that fermented GBE may have the potential to treat AD. However, there are still several points to be improved in the article : the sample should have an expert and Latin ; fermentation methods should have literature sources ; most of the reagents such as anesthetics should not indicate the batch number and manufacturer, such as the anesthetics in 2.11 did not indicate the manufacturer and batch number ; the grade and certificate number of mice should be provided ; provide animal center license number ; the design method of animal experiment is not detailed enough, and the method should be grouped according to the number of mice in each group. 3.1 What is the name of the substance corresponding to each peak in the HPLC results ? Should be improved ; because of continuous oral administration for 6 weeks, the body weight should be weighed once a week, and the dosage should be adjusted according to the body weight. The number of samples should be provided under the chart of each result, which is the basis of statistics ; the conclusion should increase the discussion of the results ; the reference format should be unified, mainly based on the literature of the past three years. Due to more problems, it is recommended to return the manuscript.

In this paper, the ginsenoside content and physiological characteristics of fermented ginseng berry extract ( GBE ) were studied. Lactobacillus plantarum was used to ferment fresh ginseng C.A. Meyer berries. The experimental results showed that the contents of Rg3 ( S ), Rg3 ( R ), Rk1, Rg5, total polyphenols and flavonoids in fermented GBE were significantly increased. It can also inhibit the cytotoxicity induced by af64a and up-regulate the expression of CHT, ChAT, VAChT, nAChRα5 and β2, m1AChR, NGF and BDNF genes. The conclusion indicates that fermented GBE may have the potential to treat AD. However, there are still several points to be improved in the article : the sample should have an expert and Latin ; fermentation methods should have literature sources ; most of the reagents such as anesthetics should not indicate the batch number and manufacturer, such as the anesthetics in 2.11 did not indicate the manufacturer and batch number ; the grade and certificate number of mice should be provided ; provide animal center license number ; the design method of animal experiment is not detailed enough, and the method should be grouped according to the number of mice in each group. 3.1 What is the name of the substance corresponding to each peak in the HPLC results ? Should be improved ; because of continuous oral administration for 6 weeks, the body weight should be weighed once a week, and the dosage should be adjusted according to the body weight. The number of samples should be provided under the chart of each result, which is the basis of statistics ; the conclusion should increase the discussion of the results ; the reference format should be unified, mainly based on the literature of the past three years. Due to more problems, it is recommended to return the manuscript.

Author Response

Thank you for your kind advise. I tried to my best for correcting according to your comments.

  1. The sample should have an expert and Latin.

 - Authentication of the sample described (page 2; lines 94-97).

  1. Fermentation methods should have literature sources.

- The reference to the fermentation method added (page 3; lines 108-109)

  1. Most of the reagents such as anesthetics should not indicate the batch number and manufacturer, such as the anesthetics in 2.11 did not indicate the manufacturer and batch number.

- All information including admin route, batch number, and manufacturer added (page 5; lines 236-267).

  1. The grade and certificate number of mice should be provided.

- Animal grade and certification # added (page 5; line 231)

  1. Provide the animal center license number

- We add the “animal center license #” (page 5; line 249).

  1. The design method of the animal experiment is not detailed enough, and the method should be grouped according to the number of mice in each group. 3.1 What is the name of the substance corresponding to each peak in the HPLC results? Should be improved.

- Groups and the number of mice in each group were described and further details are given in the figure caption of Fig. 8 (page 5; line 232, page 13; lines 480-482).

- More detailed information corresponding to each peak of Fig. 2 in 3.1 was described in Supplementary Table S3.

  1. Because of continuous oral administration for 6 weeks, the body weight should be weighed once a week, and the dosage should be adjusted according to the body weight. The number of samples should be provided under the chart of each result, which is the basis of statistics.

- In Supplementary Fig.S1, data were provided measuring body weight every 5 days for 6 weeks.

- As there was no weight loss for 6 weeks, we decided not to adjust the dose.

- As we performed all experiments in triplicate, the figures have been revised by adding the number of samples throughout the text.

  1. The conclusion should increase the discussion of the results.

- As suggested, the discussion was increased based on the results and concluded with the conclusion.

  1. The reference format should be unified, mainly based on the literature of the past three years. Due to more problems, it is recommended to return the manuscript. 

-  References were double-checked and revised in accordance with Author’s guidelines.

Reviewer 3 Report

In this submitted article, Yoon et al investigated the effects of fermented ginseng berry extract (GBE) on Alzheimer's disease (AD). The authors found that fermentation increased the content of bioactive compounds, including Rg3, Rg5, Rk1, total polyphenols, and flavonoids. In both in vitro and in vivo experiments, post-fermented GBE protected against AF64A-induced cholinergic system damage, by upregulating the expression of cholinergic genes, such as NGF and BDNF. The study further showed post-fermented GBE reduced inflammation, increased antioxidant effects, decreased amyloid-beta levels, and improved behavioral score in mouse model. Overall, the data are very solid, and the reported findings are meaningful for the field.

Author Response

Thank you for your kind advise

Reviewer 4 Report

The present study related to Improvement of Cognitive Function by Fermented Panax ginseng C.A. Meyer Berries Extracts in an AF64A-induced Memory Deficit Model is interesting. However, edits must be made before being considered for proceed in Nutrients.

Edits and suggestions :

1.     In the abstract, please specify no “curative” treatment, because there are therapeutic and preventive treatments in AD.

2.     In the keywords you could add “animal model” or “mice”.

3.     It would be interesting to briefly explain the effect of GBE on the central nervous system and please add articles line 75-77.

4.     You do not specify the number of mice in your study. Moreover, how are the mice distributed in the different groups : 100, 200 and 400 mg/kg GBE or 3 mg/kg donepezil. Control group ? N=  in each group ?

5.     Please specify the oral administration of GBE and donepezil : in food, in water, by gavage ?

6.     Instead of Figure 1, it would be of interesting for readers if you produced a schematic figure of your experimental design over the 9 weeks.

7.     Please insert the figure 4 in 3.3 section.

8.     Figure 7C and D could be improved to be more visible.

9.     Figure 8A and B could be improved to be more visible.

10.   Could you put a scale on the images of figure 11.

11.  In the limit / perspectives part, you can also question the age of the mice, they are young mice, would you have more effects in aging mice ? You can wonder about the bioavailability of the pathway oral administration used, another route of administration would have improved bioavailability (oral route by gavage, nasal route) ? and finally what is the number of mice per group, do you have a representative group?

Please, could you add a conclusion or put part of the discussion in the conclusion part

Minor editing of English language required

Author Response

Thank you for your kind advise. I tried to my best for correcting according to your comments.

  1. In the abstract, please specify no “curative” treatment, because there are therapeutic and preventive treatments in AD.

- The “Abstract” was revised in accordance with your comments (page 1; lines 16-18).

  1. In the keywords you could add “animal model” or “mice”

- ICR mouse was added to the “Keywords” (page 1; line 35)

  1. It would be interesting to briefly explain the effect of GBE on the central nervous system and please add articles line 75-77.

- Information on the effect of GBE was described in the “Discussion and Conclusion” based on the results (page 16; lines 591-595).

- As the increase in neurotrophin and cholinergic system activity by GBE was actually the result of the study results, the content in the "Introduction" was deleted and presented in the "Discussion" based on the results (page 16; lines 606-607).

  1. You do not specify the number of mice in your study. Moreover, how are the mice distributed in the different groups: 100, 200, and 400 mg/kg GBE or 3 mg/kg donepezil? Control group? N=  in each group?

- The number of animals to be used in this study and the allocations within a group were described accordingly (page 5; lines 232).

  1. Please specify the oral administration of GBE and donepezil: in food, in water, by gavage?

- The appropriate description for the administration was added (page 5; lines 243-244).

  1. Instead of Figure 1, it would be interesting for readers if you produced a schematic figure of your experimental design over the 9 weeks.

- Thank you for your suggestion to include a schematic figure of our experimental design over the 9 weeks. While we appreciate your feedback, we believe that the current Figure 1 is sufficient to convey our study design. As you can see, Figure 1 provides a clear overview of the experimental design, including the different treatments administered to the mice over time. We believe that this figure is clear and easy to understand, and we feel that adding a new schematic figure would not add significant value to the manuscript. However, we will revise the figure legend to provide additional details on the experimental design and ensure that readers can fully understand the figure (page 6; lines 253-259).

  1. Please insert Figure 4 in the 3.3 section.

- Thank you for your comment, and Figure 4 was moved to section 3.3.

  1. Figure 7C and D could be improved to be more visible.

- Figures 7C and D were improved.

  1. Figure 8A and B could be improved to be more visible.

- Figures 8A and 8B were improved.

  1. Could you put a scale on the images of Figure 11?

- Scale bar was inserted.

  1. In the limit/perspectives part, you can also question the age of the mice, they are young mice, would you have more effects on aging mice? You can wonder about the bioavailability of the pathway oral administration used, another route of administration would have improved bioavailability (oral route by gavage, nasal route)? and finally, what is the number of mice per group, do you have a representative group?

- Thank you for your comments and suggestions. Our study was primarily focused on developing an anti-dementia medicine with fermented GBE via the oral route in a young animal model. While we did not specifically consider an old animal model in the AF64A model, we believe that our findings provide valuable insights into the potential therapeutic effects of fermented GBE on cognitive function. Regarding your questions about the age of the mice, we chose to use young mice in our study to minimize the potential confounding effects of age-related changes in cognitive function. However, we acknowledge that it would be interesting to investigate the effects of fermented GBE on aging mice in future studies. In terms of the bioavailability of the oral route of administration, we chose this route because it is a common and convenient method of drug delivery. However, we recognize that other routes of administration, such as oral gavage or nasal administration, may improve bioavailability and could be explored in future studies. Finally, we used 5 mice per group in our study, which we believe is a representative sample size for this type of experiment. We conducted statistical analyses to ensure that our results were robust and reliable. However, we acknowledge that larger sample sizes may be needed to confirm our findings and increase the statistical power of the study. We appreciate your comments and will consider your suggestions in future studies.

  1. Please, could you add a conclusion or put part of the discussion in the conclusion part?

- Yes. We added a conclusion in the “Discussion and Conclusion” at the end of the section (page 17; lines 622-626).

Round 2

Reviewer 4 Report

The authors have reviewed and responded to most of my edits and suggestions, however, some methodological elements are not detailed or understandable enough.

1.     Please specify in the keyword then in the text what corresponds to ICR.

2.     The design method of the animal experiment is not clear enough (Section 2.11 and Figure 1) :

In Section 2.11 : please detail what the 6 groups correspond to.

In the figure 1, it is not clearly indicated the six groups and their number (n=).

In the figure 1 : “six groups and treated with either fermented ginseng berry extract (GBE), donepezil, or a combination of both for a period of 9 weeks” Specify what is the combination of both ?

Please specify in the oral administration group 0 : no oral administration or a placebo ?

3.     Lines 242-243 : “After a period of 2 weeks, either a vehicle, 100, 200, and 400 mg/kg of GBE, or 2 mg/kg of donepezil were administered orally for a duration of 6 weeks.” Please specify the number of oral administration per week, 1, 5 or 7 days per week ?

4.     Lines 461-463 “The mice were then orally administered with post-fermented GBE (0, 100, 200, and 400 mg/kg) and donepezil (2 mg/kg) for 6 weeks, during which no adverse effects such as weight loss were observed” Please specify if you have identify possible changes in behavior in order to analyze the effects of a treatment, other than weight, such as physiological and behavioral parameters of mice.

5.     For the in vivo study, in each figure check that it is well noted n=

6.     Line 232, you specify "The mice were allocated to six groups (seven in each group)" but in the answer to my question 11 it is noted "Finally, we used 5 mice per group in our study", so what is the right number per group? Were there any exclusions, if so due to death or other?

Author Response

  1. Please specify in the keyword then in the text what corresponds to ICR.

à Thank you for your comment. In accordance with your advice, we have revised 'ICR' to 'ICR mice' in the 'Abstract' and made corresponding changes in the main text.

  1. The design method of the animal experiment is not clear enough (Section 2.11 and Figure 1) :

In Section 2.11 : please detail what the 6 groups correspond to.

In the figure 1, it is not clearly indicated the six groups and their number (n=).

àThank you for your comment. We apologize for any confusion in the manuscript. As stated in our response, the mice were randomly assigned to the following groups: normal control group (NC; n=7), AF64A + vehicle group (Veh; n=7), AF64A + donepezil group (donepezil; n=7), AF64A + post-fermented GBE 100 mg/kg group (GBE100; n=7), AF64A + post-fermented GBE 200 mg/kg group (GBE200; n=7), and AF64A + post-fermented GBE 400 mg/kg group (GBE400; n=7). We have also updated Figure 1 to clearly indicate the six groups and their number (n=7). We hope that this revision addresses and clarifies the design method of our animal experiment. (page 5; lines 236-240)

In the figure 1 : “six groups and treated with either fermented ginseng berry extract (GBE), donepezil, or a combination of both for a period of 9 weeks” Specify what is the combination of both ?

àThank you for your comment. The combination of both in Figure 1 refers to the group of mice that were treated with both fermented ginseng berry extract (GBE) and donepezil for a period of 9 weeks. After 2 weeks of injection of AF64A, the mice were treated with either GBE (at doses of 100, 200, or 400 mg/kg) or donepezil (at a dose of 2 mg/kg) for a period of 9 weeks. Therefore, the combination group would have received both GBE and donepezil during the 9-week treatment period. This description is added and revised, accordingly. (page 6; lines 257-263)

Please specify in the oral administration group 0 : no oral administration or a placebo ?

àThank you for your comment. It means a placebo. To clarify this, we have changed the term to GBE0 or Veh.

  1. Lines 242-243 : “After a period of 2 weeks, either a vehicle, 100, 200, and 400 mg/kg of GBE, or 2 mg/kg of donepezil were administered orally for a duration of 6 weeks.” Please specify the number of oral administration per week, 1, 5 or 7 days per week ?

àThank you for your comment. We have changed to “After a period of 2 weeks, either a vehicle (0), 100, 200, and 400 mg/kg of GBE, or 2 mg/kg of donepezil was administered orally once a day for a duration of 6 weeks.” (page 5; lines 246-248)

  1. Lines 461-463 “The mice were then orally administered with post-fermented GBE (0, 100, 200, and 400 mg/kg) and donepezil (2 mg/kg) for 6 weeks, during which no adverse effects such as weight loss were observed” Please specify if you have identify possible changes in behavior in order to analyze the effects of a treatment, other than weight, such as physiological and behavioral parameters of mice.

àThank you for your comment. We monitored the mice for changes in behavior and physiological parameters in addition to weight loss, but did not observe any significant changes that would suggest adverse effects of the treatments which led us to test cognitive tests (Morris water maze and passive avoidance test). However, we did not perform any additional tests beyond those described in the manuscript. We have added this information to the manuscript to clarify. (page 12; lines 472-473)

  1. For the in vivostudy, in each figure check that it is well noted n=

à Thank you for your kind advice, we have made the appropriate adjustments to each figure, accordingly.

  1. Line 232, you specify "The mice were allocated to six groups (seven in each group)" but in the answer to my question 11 it is noted "Finally, we used 5 mice per group in our study", so what is the right number per group? Were there any exclusions, if so due to death or other?

àThank you for bringing this to our attention. We apologize for any confusion caused by the discrepancy in our manuscript due to miscommunications among authors. To clarify, we used seven mice per group in this study. Specifically, we used seven animals in each group for the water maze and passive avoidance tests. For molecular analyses such as PCR, western blotting, and ACh concentration measurements, we used five animals in each group. For immunohistochemistry, we used two animals per group. We did not exclude any animals from the study. We have corrected the number of animals used in the study in the manuscript.